# Prevalence and Antibiotic Resistance Phenotypes of *Pseudomonas* spp. in Fresh Fish Fillets

**DOI:** 10.3390/foods12050950

**Published:** 2023-02-23

**Authors:** Nesrine Ben Mhenni, Giulia Alberghini, Valerio Giaccone, Alessandro Truant, Paolo Catellani

**Affiliations:** 1Bioresources, Integrative Biology and Valorization Lab (BIOLIVAL), Higher Institute of Biotechnology of Monastir, University of Monastir, Monastir 5000, Tunisia; 2Department of Animal Medicine, Production and Health (MAPS), School of Agricultural Sciences and Veterinary Medicine, University of Padua, 35020 Legnaro, Italy

**Keywords:** *Pseudomonas* spp., fish fillets, food microbiology, antimicrobial resistance, multi-drug resistance

## Abstract

In fresh fish products, excessive loads of *Pseudomonas* can lead to their rapid spoilage. It is wise for Food Business Operators (FBOs) to consider its presence both in whole and prepared fish products. With the current study, we aimed to quantify *Pseudomonas* spp. in fresh fillets of *Salmo salar*, *Gadus morhua* and *Pleuronectes platessa*. For all three fish species, we detected loads of presumptive *Pseudomonas* no lower than 10^4^–10^5^ cfu/g in more than 50% of the samples. We isolated 55 strains of presumptive *Pseudomonas* and carried out their biochemical identification; 67.27% of the isolates were actually *Pseudomonas*. These data confirm that fresh fish fillets are normally contaminated with *Pseudomonas* spp. and the FBOs should add it as a “process hygiene criterion” according to EC Regulation n.2073/2005. Furthermore, in food hygiene, it is worth evaluating the prevalence of antimicrobial resistance. A total of 37 *Pseudomonas* strains were tested against 15 antimicrobials, and they all were identified as being resistant to at least one antimicrobial, mainly penicillin G, ampicillin, amoxicillin, tetracycline, erythromycin, vancomycin, clindamycin and trimethoprim. As many as 76.47% of *Pseudomonas fluorescens* isolates were multi-drug resistant. Our results confirm that *Pseudomonas* is becoming increasingly resistant to antimicrobials and thus should be continuously monitored in foods.

## 1. Introduction

The microbiota of fishery products is made up of a set of microbial populations (bacteria, yeasts and/or molds), and the bacteria belonging to the Pseudomonadaceae family are one microbial population, alongside enterobacteria, coliforms or lactic acid bacteria. The genus *Pseudomonas* counts over 200 species divided into 11 subspecies. These bacteria are widely spread in the terrestrial and aquatic environment, both in fresh water and salt water. Based on currently available references, the species of *Pseudomonas* (hereafter sometimes only “*P*.”) most commonly isolated in fish are *P. fluorescens*, *P. lundensis*, *P. fragi, P. anguilliseptica* and *P. putida* [1,2,3]. The species of *Pseudomonas* that is mainly considered a human pathogen is *P. aeruginosa*, but it is not so common in food products [4].

Pseudomonads are one of the most relevant specific spoiling microorganisms (SSOs) because with their excessive proliferation, they enhance the splitting of nitrogenous compounds leading to product deterioration [5]. It is widely demonstrated that Gram-negative bacteria belonging to the Pseudomonadaceae family are one of the most active microbial populations in the production of proteolytic, lipolytic and saccharolytic enzymes (also extracellular ones). Furthermore, many species of *Pseudomonas* spp. are able to synthesize yellow, green fluorescent, blue and red pigments that can lead to the development of anomalous colors [5,6,7,8]. The discoloration and the development of unpleasant off-odors and slime coats characterize the spoilage of fresh fish, whole or portioned if they are stored in the air and even if stored at refrigeration temperatures [5,9]. In fact, Pseudomonadaceae are aerobic microorganisms and grow only in the presence of molecular oxygen. In vacuum- and CO_2_-packed stored fish, the number of *Pseudomonas* spp. is reduced [5].

Grooming and filleting operations bring additional loads of microbial community to the product being processed, with the risk of shortening the commercial life of the product itself [10].

*Pseudomonas* spp. can produce tough microbial biofilms on all surfaces they come into contact with, and these biofilms can offer shelter and protection to foodborne disease bacteria. The presence of high loads of *Pseudomonas* spp. in food or in any other substrate can favor the survival of *Listeria monocytogenes*, *Staphylococcus aureus* and enteropathogenic strains of *E. coli* (STEC) [11,12,13,14]. Due to the biofilm produced by *Pseudomonas* spp., it is not easy to eliminate the bacteria that are hosted there and, on the contrary, they are able to resist various environmental stresses, including chlorine disinfectants. Nikel and co-authors reported that *P. putida* possesses metabolic mechanisms that allow it to resist oxidative stress [15]. It is also known that Pseudomonadaceae are able to withstand a certain osmotic pressure (salinity of the medium) thanks to specific molecules present in their cytoplasm that can hook osmoprotective compounds such as choline, betaine and carnitine from the environment in which the bacteria are found, favoring the survival of species such as *P. syringae* and *P. aeruginosa* [16].

In general, it is accepted that a low number of *Pseudomonas* spp. does not pose a health problem for the consumer. Instead, Pseudomonadaceae can produce histamine and become a risk to human health if they proliferate excessively in fish that are inherently high in free histidine [17]. For this and for the reasons considered above, it is therefore wise that FBOs take into due consideration the presence of *Pseudomonas* spp. in whole fresh fish as well as in fresh fillets, formally giving to it the qualification of “process hygiene criterion” pursuant to EC Regulation n.2073/2005 [18] that establishes some “microbiological criteria for foodstuffs”.

The aforementioned Regulation is addressed to the FBOs who use it as a reference to check and validate their self-monitoring plan as well as the safety level of their products as required by European laws. The process hygiene criteria currently provided by the Regulation n.2073/2005 for fishery products are the loads of *E. coli* and coagulase-positive staphylococci; in addition, only shelled and shucked products of cooked crustaceans and shellfish are the fishery products considered.

The consumption of fishery products is regularly increasing, especially in low- and middle-income countries. Between 1961 and 2016, the demand for fish products increased by about 3.2% per year, while the need for meat from terrestrial animals increased by only 2.8% [19,20]. According to the FAO Report 2022 on “The State of World Fisheries and Aquaculture”, in recent years, there has been a sharp increase in the consumption of fresh, chilled and frozen fish products, while the consumption of salted, smoked or canned products remains stable. At the same time, aquatic food production is increasing; China, Norway, Vietnam, Chile and India are the major producers and exporters in the world [19]. In 2020, the global production of fishery products reached 178 million tons (with aquaculture representing 49.2% of the total), and it is forecasted to grow another 15% by 2030 [19].

Aquatic food systems are under growing pressure, and many reports document that the use of antimicrobials in fish farming is consequently growing. Researchers estimate that in 2030, about 13,495 tons of antimicrobials will be used in aquaculture, which, therefore, represents 5.7% of the total amount used in the world (by adding those used in humans and in all other animals) [21]. This also draws much attention to the fact that most classes of antimicrobials are used for the treatment of bacterial infections in both veterinary and human medicine [22]. This increases the speed of selection of drug-resistant bacteria [23], and antimicrobial-resistant bacteria can reduce the effectiveness of treatments, especially in Gram-negative bacterial infections [24]. Thus, the careless use of antibiotics in aquaculture is a contributing factor to the rise in antimicrobial resistance (AMR), leading to potential animal, human and ecosystem consequences [21].

The relationship between the detection of antibiotic-resistant bacteria in humans and antibiotic use in food animals continues to be a topic of discussion among researchers. Various studies have suggested that the use of antibiotics in animals can directly affect human health through direct contact with antibiotic-resistant bacteria from food animals [23]. Other authors argue indirect effects that result from contact with resistant organisms that have spread to various components of the ecosystem (e.g., water, soil, etc.), and so they can indirectly pass from animals to humans and vice versa. This indirect effect can be greatly increased by the horizontal transfer of mobile genetic elements such as conjugative plasmids, phages and transposons [25].

From a holistic view of AMR in different sectors, there are few studies that directly consider foods and not clinical isolates. Moreover, current AMR research, surveillance programs and dietary risk assessment studies (as the potential transmission of AMR), focus mainly on food-producing terrestrial animals and a few indicator bacteria. So, considering that world seafood production is increasing every year, our study aims to provide additional data to implement those already known regarding *Pseudomonas* spp. which is considered a dominant bacterial genus in food processing facilities and in the microbial community of many fish products [4].

Many strains of *Pseudomonas* have been shown to possess a high level of intrinsic resistance to most antibiotics [26]. This intrinsic resistance is mainly conferred by concurrent mechanisms, such as low outer membrane permeability, efflux systems that pump antibiotics out of the cell and the production of antibiotic-inactivating enzymes such as β-lactamases [17,27]. For now, various resistance genes have been described in the literature, and as it is a possible source of new resistance genes, *Pseudomonas* spp. should be better monitored [28] in their evolution.

Considering only the strains of *Pseudomonas* spp. isolated from food, previous studies reported in the literature confirm that various strains are able to resist some antimicrobial agents of different classes, especially β-lactams such as penicillins, cephalosporins, carbapenems, and monobactams [29,30]. For example, the study reported by Kačániová and co-authors [31] reveals a high proportion of resistant strains among the *Pseudomonas* spp. isolated originated from fish. Moreover, all *Pseudomonas* were resistant to meropenem. According to Fazeli and Momtaz [32], bacterial strains exhibited the highest level of resistance to penicillin (100%) followed by tetracycline (90.19%), streptomycin (64.70%) and erythromycin (43.13%). Shabana and co-authors [33] showed that *P. fluorescens* strains were resistant to more than two classes of antibiotics and, therefore, they are considered multidrug-resistant (MDR) strains and accounted for 29.7%.

The aim of our study was to determine and characterize a *Pseudomonas* spp. population in fresh fish fillets at the beginning of their shelf life and evaluate the degree of antimicrobial resistance of the isolated strains in order to provide useful data for monitoring its development from the point of view of food safety.

## 2. Materials and Methods

### 2.1. Samples Collection

A total of 75 fish fillets intended for large retailers were collected from a fish industry located in the Venetian area (Italy) in 5 different sampling times from March to May 2022. Each time, 15 samples were analyzed, namely:Five salmon (*Salmo salar*) fillets reared in Norway;Five plaice (*Pleuronectes platessa*) fillets caught using trawling in the Northeast Atlantic Ocean;Five northern cod (*Gadus morhua*) fillets caught using longlining in the Northeast Atlantic Ocean.

The fish arrived in Italy by ship and under melting ice or frozen, as established by the EC Regulation n.853/2004. The samples of salmon and plaice were fresh, while the cod fillets were defrosted at the processing site. The fillets were collected from the fish industry early in the morning after packaging and immediately transferred to the laboratory. During transport (less than one hour), they were preserved at refrigeration temperature in expanded polystyrene containers with eutectic plates.

In the laboratory, the fillets were immediately subjected to microbiological analysis for Total Viable Count and *Pseudomonas* spp. count using the following methods: UNI EN ISO 6887-1: 2017, UNI EN ISO 7218: 2013, UNI EN ISO 4833-1: 2013, UNI EN ISO 11133: 2014, UNI EN ISO 13720:2010.

### 2.2. Samples Processing

According to the guidelines provided by UNI EN ISO 6887-1:2017, 10 g of each sample was moved into a sterilized container with 90 mL of Buffered Peptone Water (BPW) under strict hygienic measures, and homogenization was performed using a Stomacher 400 Lab blender.

From this mixture, 1 mL was transferred to a sterilized test tube containing 9 mL of BPW, from which ten-fold serial dilutions were processed up to 10^−5^. The prepared samples were subjected to the determination of *Pseudomonas* counts using *Pseudomonas* Agar Base (PAB) supplemented with CFC Supplement (Liofilchem^®^) and Total Viable Count (TVC) using Plate Count Agar (Merck^®^). The *Pseudomonas* Agar Base inoculated plates were incubated in a fridge-thermostat at 25 °C for 48 h, while the Plate Count Agar inoculated plates were incubated at 31 °C for 48–72 h.

### 2.3. Plate Counting and Statistical Analysis

The colonies were counted using the UNI EN ISO 7218:2013 standard as a reference. The results were expressed in the colony-forming unit (cfu)/g and then subjected to the determination of arithmetic mean and statistical analysis.

Log-transformed data (Total Viable Count and *Pseudomonas* spp.) were analyzed using a linear ANOVA model that included the fixed effects of fish type and sample ID (nested within the fish type). Least squares means (ls-means) were calculated and post hoc pairwise comparisons were performed using the Bonferroni correction. Data expressed as an index [formula: number of figures-1; 10^ (order of magnitude with powers of 10)] were submitted to non-parametric analysis (Kruskall–Wallis test) to assess the effect of the type of fish.

Post hoc pairwise comparisons were estimated using the Steel–Dwass–Critchlow–Flinger correction. *p* < 0.05 was considered significant. All the analyses were carried out with SAS (SAS Institute Inc, Cary, NC, USA 2017) and XLStat (XLSTAT statistical and data analysis solution. New York, NY, USA).

### 2.4. Isolation of Pseudomonas *spp.* and Species Identification

At least 5 colonies grown on PAB were selected from each plate and their oxidase and catalase activities were tested. The oxidase test was performed by smearing a fresh colony of each isolate onto a sterile oxidase filter paper disc soaked in distilled water. The appearance of purple color indicates a positive result of oxidase activity. The catalase test was carried out using hydrogen peroxide and the positive reaction was indicated by immediate bubbling.

At this stage, if the colonies gave positive results to the two tests mentioned, the non-fermentation activity was evaluated using the Kligler Iron test. In the Kligler Iron Agar infixion test, bacterial growth was evaluated after 24 h of incubation at 37 °C (time and temperature are those written in the technical data sheet of the media). Presumptive *Pseudomonas* spp. developed only on the surface because they are strictly aerobic bacteria. Furthermore, this bacterium does not ferment glucose or lactose but can split peptones, so the indicator turned purple (alkaline pH) on the surface.

Finally, the counts were corrected proportionally to the results of the tests provided by the ISO mentioned above.

Some of the colonies which proved positive for the 3 tests were identified by a unique alphanumeric code, purified on Brain Heart Infusion agar (Thermo Scientific™ Oxoid™), inoculated into Plate Count Agar tubes and stored in the refrigerator at 4 °C. Then, the isolated colonies (*n* = 55) were subjected to biochemical identification with the BIOLOG^®^ system. Biolog’s carbon source utilization technology identifies microorganisms depending on their characteristic metabolic pattern produced as a consequence of some pre-selected test. The reading is performed by comparing the results to an extensive database.

### 2.5. Determination of Pseudomonas *spp.* Phenotypic Antibiotic Resistance

The analyses were performed following the recommendation of EUCAST 2022 for the method of execution [34]. Antimicrobial resistance was tested using the standard disc diffusion method.

Suspensions of *Pseudomonas* spp. were cultivated on Mueller Hinton agar (Merck^®^) with modifications regarding incubation temperature for the psychrotrophy of the bacterium itself. Then, antimicrobial discs (Liofilchem^®^) were placed on the agar surface taking care to preserve as much sterility as possible. All the strains of *Pseudomonas* spp. were tested against 15 different commercial antimicrobial agents chosen among the most used and important in human medicine and aquaculture worldwide [21,35,36]; the concentrations were instead chosen on the basis of preliminary studies. The antimicrobials were ciprofloxacin (5 μg/disk), enrofloxacin (5 μg/disk), tetracycline (30 μg/disk), rifampicin (30 μg/disk), erythromycin (15 μg/disk), vancomycin (30 μg/disk), clindamycin (10 μg/disk), meropenem (10 μg/disk), trimethoprim (5 μg/disk), penicillin G (10 IU/disk), ampicillin (10 μg/disk), amoxicillin (30 μg/disk), flumequine (30 μg/disk), florfenicol (30 μg/disk) and sulfadiazine (300 μg/disk). Inoculated plates were incubated at 31 °C for 24 h. Then, the diameters were measured in millimeters with a caliper.

For the interpretation of the results, neither EUCAST nor CLSI (Clinical and Laboratory Standards Institute) provides criteria and reference values for *Pseudomonas* spp., so, it was decided to consider as “highly resistant” those cases in which the diameter of the inhibition zone was ≤10 mm.

## 3. Results and Discussion

### 3.1. Pseudomonas *spp.* Count, Total Viable Count, Data Elaboration

The load of *Pseudomonas* spp. in the samples analyzed has been reported in Figure A1. Indeed, in the salmon fillets, in 19 samples out of 25 (76%), the initial loads were between 10^4^ and 10^5^ cfu/g, a rather significant load considering that these are fresh fish products. In the remaining samples, the charges of *Pseudomonas* spp. were between 10^3^ and 10^4^ cfu/g. In only one sample, 10^6^ cfu/g has been detected.

In the plaice fillets, 13 out of 25 (52%) of the samples analyzed showed a load of *Pseudomonas* spp. between 10^4^ and 10^5^ cfu/g, and the remaining 12 samples showed loads between 10^5^ and 10^6^ cfu/g since the beginning of their shelf life.

According to Silbande and co-authors [37], *Pseudomonas* loads equal to or greater than 10^6^ cfu/g may already be sufficient to compromise the organoleptic qualities of fresh fish products.

Therefore, in the salmon and plaice fillets, the loads of *Pseudomonas* spp. were, on average, between 10^4^ and 10^5^ cfu/g; nevertheless, some of the samples presented an initial load between 10^5^ and 10^6^ cfu/g of the fillet. Higher loads in some fillets could be due to transport or to the processing environment and also to the moment of processing (beginning vs. end) of that specific product compared to the other products belonging to the same batch.

In the defrosted northern cod fillets, the number of samples that contained 10^4^–10^5^ cfu/g was 14 out of 25 (56%), while the number of samples that showed amounts of *Pseudomonas* spp. between 10^5^ and 10^6^ cfu/g was 10. In only 1 sample of northern cod fillet, an exceptionally low load of *Pseudomonas* spp. (i.e., 10^2^ cfu/g) was detected. In addition to the reasons described above for salmon and plaice, the remarkable loads of cod fillets could be also explained by the fact that cod fillets were a defrosted product, so the load of *Pseudomonas* spp. could have been affected by the specific pre-existing microbial population in the fish prior to freezing, although it is admitted that freezing may have inactivated part of this specific microbial population [38].

It is also interesting to note that loads of *Pseudomonas* spp. in plaice and cod fillets have progressively increased with the progress of the season and, therefore, going towards higher environmental temperatures. Indeed, the loads of *Pseudomonas* spp. were relatively low in the samples analyzed in March and April, while they increased in the April/May period [39]. However, this was not observed in salmon fillets that presented a rather “homogeneous” microbial distribution and, in any case, further and larger studies should be performed to confirm this hypothesis.

Generally, the average for the counts of pseudomonads detected in all three species analyzed is around 10^4^ cfu/g (see Table 1) with some occasional increases towards higher loads (10^5^–10^6^ cfu/g) rather than lower levels.

Regarding the Total Viable Count (TVC) in the three species of fillets analyzed (see Figure A2), the values were in most cases comprised between 10^4^ and 10^5^ cfu/g or between 10^5^ and 10^6^ cfu/g. In 10 samples out of 75, the Total Viable Count was higher, between 10^6^ and 10^7^ cfu/g. The arithmetic mean values were around 10^5^ cfu/g (see Table 1).

These values lead to the conclusion that a large part of the microbiota found in the analyses was made up of Pseudomonadaceae with other minor microbial populations. This confirms that the loads of *Pseudomonas* spp. have an important influence on the shelf-life of the fish fillets considered.

The ANOVA model using log-transformed Total Viable Count data showed a significant effect of the fish type (*p* < 0.001). Back transformed ls-means pointed out that the loads of *Pseudomonas* spp. in salmon fillets were significantly lower than those in cod and, respectively, in plaice fillets (1.07 × 10^5^ vs. 2.57 × 10^5^ vs. 2.56 × 10^5^ cfu/g, *p* < 0.01, see Table 1).

The effect of fish type was also significant for *Pseudomonas* spp. loads, with salmon being significantly lower than plaice (2.33 × 10^4^ vs. 6.71 × 10^4^ cfu/g, *p* = 0.028, Table 1) and cod being between the two above (4.67 × 10^4^ cfu/g). The Kruskall–Wallis test showed a significant effect of fish type only for *Pseudomonas* spp. (*p* = 0.021) with the same results as the previous linear ANOVA model (Figure 1).

### 3.2. Phenotypic Identification of Pseudomonas Species

During our study, from the analyzed fish fillets, we selected 55 strains of presumptive *Pseudomonas* spp. that have been biochemically identified with the BIOLOG^®^ system. Overall, 67.27% (37 up to 55) of the strains were definitively confirmed as *Pseudomonas* spp., while the other genera were all Gram-negative bacteria, such as *Kluyvera*, *Proteus*, *Citrobacter*, *Klebsiella*, *Serratia*, *Roseomonas* and *Providencia*. This finding is very similar to the one of another recent study carried out on the salmon processing environment where, out of 222 presumptive *Pseudomonas* isolates, 68% were confirmed as *Pseudomonas* [40]. This means that the identification of species on a certain number of bacterial colonies isolated is necessary to establish a higher degree of precision in the loads recorded. The bias of this methodological approach is represented by the economic costs and waiting times to obtain the final results.

*Pseudomonas* isolates originating from salmon, cod and plaice fillets are diverse with many species represented. Our isolated *Pseudomonas* species were *P. fluorescens*, *P. fragi*, *P. lundensis*, *P. marginalis*, *P. syringae*, *P. taetrolens*, *P. chlororaphis*, *P. tolaasii* and *P. viridilivida*. Indeed, 45.95% of the strains are represented by *P. fluorescens*, which largely dominates the Pseudomonadaceae population in the analyzed fillets, followed by *P. fragi* (21.62%), *P. marginalis* (10.81%), *P. taetrolens* (8.11%), *P. lundensis, P. syringae, P. tolaasii, P. viridilivida* and *P. chlororaphis* (2.70% each one).

It should be noted that this study has some limitations, mainly the lack of molecular identification using 16S rRNA and some housekeeping genes. It is, however, true that the genus *Pseudomonas* is very large and includes several hundred unclassified strains. So, for this complex, genus sequencing of the 16S rRNA gene can often only identify the three main lineages (*P. aeruginosa*, *P. pertucinogena* and *P. fluorescens*) but cannot give information about the species [40].

These data confirm what has already been described in the literature, which indicates that among the major components of Pseudomonadaceae in fishery products, there are *P. fluorescens* and *P. fragi* [1,2,3]. It should be noted that *P. aeruginosa* was never found among the strains isolated, and this is in line with what is reported by other studies [4] which highlight a low prevalence of the *P. aeruginosa* species in foods, including in drinking water.

As it is known, *P. fluorescens* and *P. marginalis* were able to produce fluorescent pigments on culture medium when illuminated with a Wood’s lamp light (Figure A3). Additionally, some strains of *P. chlororaphis* also produced fluorescent pigments.

### 3.3. Antimicrobial Resistance

In order to ensure the best growing conditions, in our study we changed the temperature of incubation of the Mueller–Hinton plates as the tested isolates were psychrophiles and could not grow well at high temperatures. The assay for these was conducted at 31 °C for 24 h. It was decided to consider as “highly resistant” those cases in which the diameter of the inhibition zone was ≤10 mm as it was impossible to find guidelines that provided identification criteria. The reasons were that, on the one hand, the incubation temperature was different and, on the other hand, neither EUCAST nor CLSI give enough information about the *Pseudomonas* species.

Resistance was found with a prevalence greater than 50% (see Figure A4) in the following cases: penicillin G (penicillins), ampicillin (penicillins), amoxicillin (penicillins), tetracycline (tetracyclines), erythromycin (macrolides), vancomycin (glycopeptides), clindamycin (lincosamides) and trimethoprim (diaminopyrimidines).

It should be noted that, according to the WHO classification (2019) [36], macrolides and glycopeptides are CIAs (Critically Important Antimicrobials), while lincosamides and diaminopyrimidines are HIAs (Highly Important Antimicrobials).

Unlike Kačániová and co-authors [31], who found resistance of all *Pseudomonas* strains to meropenem, in our study, only 16.22% of the isolates were resistant to this antibiotic. Instead, according to Fazeli and Momtaz [32], the bacterial strains showed a high level of resistance to penicillin (86.49%), tetracycline and erythromycin but with greater resistance to erythromycin (64.86%) rather than to tetracycline (54.05%). In our case, these two were surpassed (or equaled) by clindamycin (70.27%), vancomycin and trimethoprim (both 64.86%).

In the literature, various resistance mechanisms are described that could explain some of the cases detected in our study, such as β-lactamases against penicillins, efflux pump against erythromycin and dihydrofolate reductase against trimethoprim [24,27,28,41].

Countless acquired resistance genes have also been discovered [28]; however, few studies on the possible mechanisms of the resistance owned by *Pseudomonas* spp. against clindamycin and vancomycin are reported in the scientific literature.

In our research, we noticed that 76.47% of the strains of *P. fluorescens* were resistant to more than two antibiotic classes and so we considered them as multi-drug resistant (MDR) according to Shabana and co-authors’ definition of “MDR strains” [33].

## 4. Conclusions

Based on the data obtained and the statistical analyses mentioned above, we can conclude that, under the basic hygienic conditions of production established by the EC Regulations n.852/2004 and 853/2004, fresh fish fillets (salmon, plaice, cod) intended for large-scale retailers have a load of *Pseudomonas* spp. of about, on average, 10^4^ cfu/g. In salmon fillets, the counts of *Pseudomonas* spp. recorded have sporadically reached even 10^5^ and 10^6^ cfu/g. In contrast, in plaice and cod fillets, the upward swing of *Pseudomonas* loads to between 10^5^ and 10^6^ cfu/g were more frequent.

Moreover, in our study, we noted a significant effect of the fish type on the *Pseudomonas* population of the fillets. The loads of *Pseudomonas* spp. in salmon fillets were significantly lower than in plaice fillets (*p* = 0.028), while in cod fillets, they ranged between the two above mentioned species.

Considering the Total Viable Count of around 10^5^ cfu/g (mean value), we can also conclude that a large part of the microbial community found on the analyzed fish fillets was made up of Pseudomonadaceae with other minor microbial populations. The most frequently isolated *Pseudomonas* species was *P. fluorescens.*

As *Pseudomonas* is the main SSO of fresh fish fillets, it is certainly worth keeping an eye on the colony-forming units per gram of fish muscle. In our opinion, fresh fish fillets should contain low loads of *Pseudomonas* spp. during production in order to avoid rapid degradation of the product during its shelf life. It would be advisable that in the fresh fish fillets, the initial *Pseudomonas* load should be lower than 10^4^ or 10^5^ cfu/g to not exceed the concretely spoiling loads of these bacteria. In general, in the literature [37,42], it is admitted that up to 10^6^ cfu/g of SSO, the organoleptic quality is fine. Furthermore, considering the normal growth potential of *Pseudomonas* spp. in refrigerated fresh fish products, it is possible to act both on conservation techniques (for example vacuum packing) and on greater hygiene in the fish processing environment to limit their proliferation. It would be advisable that, at the refrigeration temperature, fresh fish fillets have a shelf life of no more than six days; five days would be even better.

Moreover, our antimicrobial resistance study on the 37 isolated strains of *Pseudomonas* spp. showed that these bacteria have a high level of phenotypic resistance to various antimicrobial classes. More specifically, 8 out of 15 antimicrobials tested proved ineffective in more than 50% of the strains. These eight antimicrobials are penicillin G 10 IU, ampicillin 10 μg, amoxicillin 30 μg, tetracycline 30 μg, erythromycin 15 μg, vancomycin 30 μg, clindamycin 10 μg and trimethoprim 5 μg. This highlights that 76.47% of *P. fluorescens* strains were multi-drug resistant.

In the literature, various resistance mechanisms are described (both intrinsic and acquired) such as low outer membrane permeability, β-lactamases synthesis and efflux pump systems. However, it is worth noting that few studies on the possible mechanisms of the resistance of *Pseudomonas* spp. against clindamycin and vancomycin are available. In order to deepen our research and obtain more relevant results, our recommendation is to complete this study with a genomic analysis by performing a whole genome sequencing of the isolated strains of *Pseudomonas* (WGS).

Therefore, our results suggest that *Pseudomonas* spp. should be monitored as possible source of other resistance genes. It would be preferable to complete this not only on strains isolated from clinical specimens but also on the strains isolated from food samples.

## Figures and Tables

**Figure 1 foods-12-00950-f001:**
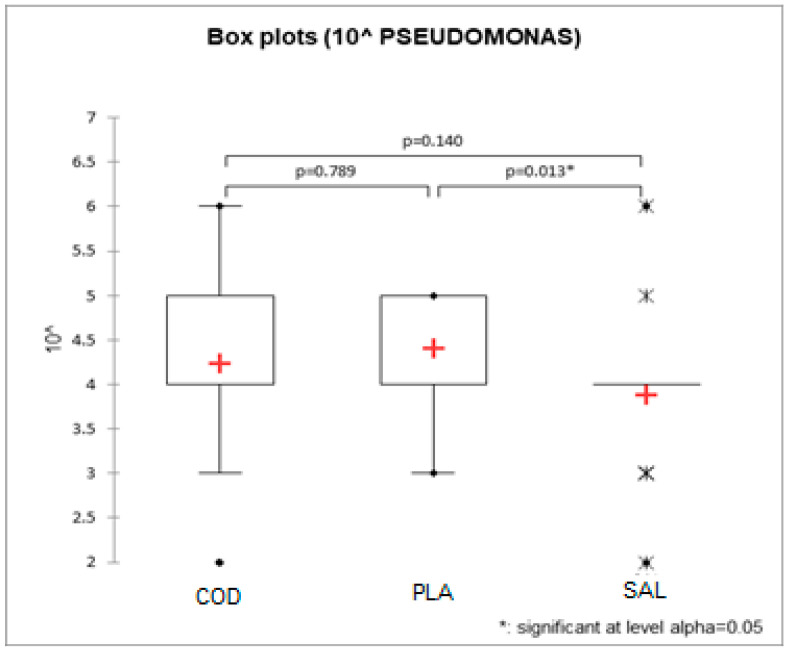
Box plot of the 10^ index (order of magnitude with powers of 10) for *Pseudomonas* spp. The boxes contain 50% of the data distribution, the red crosses represent the average values, the whiskers are proportional to the interquartile range and the points and stars are the outliers as a function of the different interquartile ranges.“COD” refers to northern cod, “PLA” to plaice and “SAL” to salmon.

**Table 1 foods-12-00950-t001:** TVC and *Pseudomonas* spp. back-transformed ls-means with 95% CI in the three species of fish fillets.

Fish Fillets	TVC Mean Value (cfu/g)	*Pseudomonas* spp. Mean Value (cfu/g)
*Gadus morhua*	2.57 × 10^5^ (1.83–3.61 × 10^5^) ^a^	4.67 × 10^4^ (2.68–8.16 × 10^4^) ^a,b^
*Pleuronectes platessa*	2.56 × 10^5^ (1.82–3.60 × 10^5^) ^a^	6.71 × 10^4^ (3.85–1.17 × 10^4^) ^a^
*Salmo salar*	1.07 × 10^5^ (7.65–1.51 × 10^5^) ^b^	2.33 × 10^4^ (1.34–4.06 × 10^4^) ^b^

^a,b^ Different letters along columns mean significantly different values (*p* < 0.05).

## Data Availability

Not applicable.

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
