# Peer review of "Prevalence and Antibiotic Resistance Phenotypes of Pseudomonas spp. in Fresh Fish Fillets"

_foods, 2023, doi:10.3390/foods12050950_

Round 1
Reviewer 1 Report
The study entitled “Prevalence and Antibiotic Resistance Phenotypes of Pseudomonas spp. in fresh fish fillets” focused on the very important issue associated with quality of fresh fish fillet that are common food of many people all over the world. One of the main indicators affecting on quality of fish fillet is bacterial load. In the present paper the authors have combined different approaches in order to estimate the bacterial loads on fillets of three economically valuable fish species. The introduction section of the present paper is well written. But in the same time, I want to point out several notes:
The authors show that different groups of bacteria may be present on the fillets of different fish species. At the same time, the authors write that the presence of these bacteria is not new or unique finding. In my opinion, in order for the article to become really interesting, the authors need to answer the question: "why is such a bacterial load observed?". To do this, it is necessary to indicate more data on the conditions for catching fish, storage conditions on a ship, transportation from Norway and North-East Atlantic Ocean to Italy, and other conditions that potentially could be associated with bacterial load. Without discussing this issue, this study does not bring anything really new. Except that the people in the Venetian area now know that in their fish industry, the fillets of salmon, plaice, and northern cod have such a bacterial load and some of these bacteria are resistant to one or several antibiotics.
The section “Results and Discussion” is actually “Results” because of majority of this part is sentences described the obtained results with some authors’ comments but without real discussion. For example, in subsections 3.1 and 3.2 there is only one reference in each subsection. I understand that the number of references is not a main criterion but 7 references for all discussion part, I guess, is not enough for such important topic as bacterial load of fish products. The "Conclusion" section also describes the obtained results but in slightly different words.
Line 336: “…in the normal conditions of production, fresh fish fillets (salmon, plaice, cod)…”
Note: in M&M section there was no information described what is “normal conditions of production”.
Lines 336-338: “…in the normal conditions of production, fresh fish fillets (salmon, plaice, cod) intended for large-scale retailers, have load of Pseudomonas spp. of about, on average, 104 cfu/g”.
Note: Why did the authors this conclusion? Half of analyzed samples of plaice and cod had 105-106 cfu/g. I understand that arithmetically, when calculating the average for the studied fish, the result was 104 cfu/g. But why did the authors unite three different fish species from two different sample points (Norway and North-East Atlantic Ocean)? Probably the different levels of bacterial load were associated with various places where these fish were collected but it did not discuss in the present paper.
Line 348: “…fish fillets should be fixed at 106 cfu/g of food at the beginning of the shelf life”.
Note: I agree that at such bacterial load the fish fille looks fresh. But have other tests been done to conclude that these fish fillets with 106 cfu/g are safe for human health before recommend such cfu/g level as a normal?
Lines 354-356: “It would be advisable that fresh fish fillets have a shelf life of no more than six days; five days would be even better”.
Note: at what temperature?
Author Response
Dear Reviewer, we have considered Your observations and we have made our modifications and integrtions directly in the new version of our article. See the attached file.

Reviewer 2 Report
Dear Authors
The spoilage microbiota is very important for a food product. In this article you have studied Pseudomonas, a very important spoilage bacteria, and the characteristics of this genera. In this era of genomics I think it is interesting with a paper based on biochemical methods. DNA is a powerful molecule, but since you are not analysing DNA you should not mention it in the paper. OK, maybe at the end when summing up but not in these lines (line 186 and 211). Use it in the conclusion.
line 27 Reformulate the first sentence.
line 34 Ref 4 is twice in the ref. list (also no 25) remove one of them
line 63 I have not seen the word charges used in this context. (e. g. quantities?) Change this word throughout the text.
line 78, 73 fix error?
line 115 Why focus on resistance genes in the introduction when no results in this paper. ? (remove table 1)
line 134 add sampling before times
line154 mL not ml (change all over)
Remember latin names in italic
line 184 more detailed description on the use of the Biolog system.
log transformed values (table 2, 3, 4)
from line 289 to 293 (latin names =italics?)
line 302 and further. Is it possible to get a figure of which P. species is resistant to what antimicrobial? Dose /disk can be omitted (in MM)
Author Response
Dear Reviewer, we have considered Your observations and we have made our modifications and integrations directly in the new version of our article. See the attached file.

Round 2
Reviewer 2 Report
Dear Authors,
The manuscript is improved, but there is still some correction to be made.
line 30 exchange "in specific" with "one"
line 30 add "acid" between lactic and bacteria
line 73 + I do not agree with exchange of numbers in the EC regulation, see https://eur-lex.europa.eu/legal-content/EN/TXT/PDF/?uri=CELEX:32005R2073&from=EN. Additionally you have to add it as a reference.
Success to your work
Author Response
We have modified our article according to the comments of Reviewer 2, See attached file.
